# Oncotype DX Test Receipt among Latina/Hispanic Women with Early Invasive Breast Cancer in New Jersey: A Registry-Based Study

**DOI:** 10.3390/ijerph18105116

**Published:** 2021-05-12

**Authors:** Nicholas Acuna, Jesse J. Plascak, Jennifer Tsui, Antoinette M. Stroup, Adana A. M. Llanos

**Affiliations:** 1Department of Preventive Medicine, Keck School of Medicine, University of Southern California, Los Angeles, CA 90032, USA; nacuna@usc.edu (N.A.); tsuijenn@usc.edu (J.T.); 2Department of Internal Medicine, College of Medicine, The Ohio State University, Columbus, OH 43210, USA; jesse.plascak@osumc.edu; 3Department of Biostatistics & Epidemiology, Rutgers School of Public Health, Piscataway, NJ 08854, USA; ams722@sph.rutgers.edu; 4Rutgers Cancer Institute of New Jersey, New Brunswick, NJ 08901, USA; 5New Jersey State Cancer Registry, New Jersey Department of Health, Trenton, NJ 08625, USA

**Keywords:** breast cancer, Oncotype DX^®^, Latina/Hispanic women, recurrence risk scores, 21-gene assay

## Abstract

Oncotype DX^®^ (ODX) is a valid test of breast cancer (BC) recurrence risk and chemotherapy benefit. The purpose of this study was to examine prevalence of and factors associated with receipt of ODX testing among eligible Latinas/Hispanics diagnosed with BC. Sociodemographic and tumor data of BC cases diagnosed between 2008 and 2017 among Latina/Hispanic women (*n* = 5777) were from the New Jersey State Cancer Registry (NJSCR). Eligibility for ODX testing were based on National Comprehensive Cancer Network guidelines. Multivariable logistic regression models of ODX receipt among eligible women were used to estimate adjusted odds ratios (AOR) and 95% confidence intervals (CI) by demographic and clinicopathologic factors. One-third of Latinas/Hispanics diagnosed with BC were eligible for ODX testing. Among the eligible, 60.9% received ODX testing. Older age (AOR 0.08, 95% CI: 0.04, 0.14), low area-level SES (AOR 0.58, 95% CI: 0.42, 0.52), and being uninsured (AOR 0.58, 95% CI: 0.39, 0.86) were associated with lower odds of ODX testing. While there was relatively high ODX testing among eligible Latina/Hispanic women with BC in New Jersey, our findings suggest that age, insurance status, and area-level SES contribute to unequal access to genetic testing in this group, which might impact BC outcomes.

## 1. Introduction

In the United States (US) an estimated 281,550 new BC cancer cases and 43,600 BC cancer deaths will occur in 2021 [1]. In the state of New Jersey, breast cancer (BC) is the second leading cause of death among women of all races/ethnicities, but the leading cause of cancer death among Latina/Hispanic women [2].

Genetic expression profiling (GEP) has been a useful mechanism allowing for the use of DNA microarrays to view expression of genes within potentially cancerous cells [3]. While several GEPs have been developed, Oncotype DX^®^ (ODX) is one test that is used to inform treatment decisions and is a strong predictor of BC-specific mortality [4]. Those with a recurrence score (RS) categorized as low (<18) have the lowest 5-year BC-specific mortality among both lymph node-negative and lymph node-positive cases when compared to those with an intermediate (18–30) or high (≥31) RS [4].

Utilization of ODX has been recommended by the National Comprehensive Cancer Network (NCCN), dating back to 2008, for women diagnosed with breast tumors that are hormone receptor-positive (HR+), human epidermal growth factor 2-negative (HER2−), lymph node-negative, Stage I or II, and >5 millimeters (mm) [5,6,7]. Beginning in 2015, NCCN guidelines included lymph node-positive cases and any tumor size in the eligibility for ODX testing. A nationwide retrospective cohort study, utilizing the National Cancer Database (NCDB), found that only one-third of eligible BC cases received the test [8]. Moreover, eligible Latina/Hispanic women diagnosed with BC have significantly lower odds of being guideline-concordant (i.e., ODX test receipt) compared to non-Hispanic women diagnosed with BC [8,9,10]. A recent study showed that Black women were more likely to have higher BC-specific mortality compared to Non-Hispanic White women with similar ODX RS, suggesting the importance of investigating ODX testing among racial/ethnic minority populations, such as Latina/Hispanic women [11]. Ethnic subgroups and nativity might be associated with ODX test receipt based on prior evidence of differences in cancer mortality [12], which may be due to differences in treatment owing to ODX testing differences. Other factors that have been shown to be strong predictors of ODX testing among eligible BC cases include tumor size, grade, socioeconomic status, cancer stage, age at diagnosis, and insurance status [4,9,10,13].

Evidence suggests that ODX testing increases confidence in treatment decisions among physicians and patients [14], which is important as many racial/ethnic minorities in the US have reduced access to standards of care relative to non-Hispanic white (NHW) individuals [15,16]. Moreover, data show that Latina/Hispanic women are more likely to accept adjuvant chemotherapy if recommended by their clinician [17]. This suggests that ODX test receipt may increase the likelihood of appropriate treatment in this group as clinicians can make recommendations based on ODX results.

In this study, we examined the prevalence of ODX test receipt among eligible Latina/Hispanic women diagnosed with BC in the state of New Jersey between 2008 to 2015, and more specifically, the factors associated with under-utilization of ODX testing among eligible women. Our *a priori* hypothesis was that lack of ODX testing among eligible Latina/Hispanic women diagnosed with BC is associated with older age at diagnosis, lower area-level socioeconomic status (SES), having no insurance, birthplace outside of the US, Hispanic ethnic subgroup, and higher cancer stage, tumor size, and tumor grade. We also performed exploratory analysis to understand the distribution of ODX RS among all Latina/Hispanic BC patients to understand the extent to which BC patients in this group are receiving ODX tests even when they are ineligible.

## 2. Methods

### 2.1. Study Population and Data Collection

Population-based BC data were provided by the New Jersey State Cancer Registry (NJSCR), including women diagnosed from 1 January 2008 through 31 December 2017. In this study, “Latina/Hispanic” refers to individuals classified as Hispanic (i.e., originating or descending from Spanish-speaking countries, including Spain and all nations from Latin America except Brazil [18]) and Latina (i.e., originating or descending from Latin American countries—South and Central America, including Brazil) [19]. Of the total sample of 57,150 BC cases, 5788 (9.9%) were classified as Latina/Hispanic based on standardized data promulgated by the North American Association of Central Cancer Registries (NAACCR). Ethnicity data were derived from a combination of Spanish/Hispanic Origin (NAACCR #190) submitted to NJSCR by reporting facilities and the NAACCR Hispanic Identification Algorithm (NHIA) [20]. Direct identification through Spanish/Hispanic Origin was used to classify cancer cases as Mexican (including Chicano), Puerto Rican, Cuban, South or Central American (except Brazilian), “other” Spanish/Hispanic origin, and Dominican. Indirect identification through the NAACCR NHIA algorithm relies heavily on matched surnames. The NHIA classification has been shown to have high levels of sensitivity (84.37%—proportion of self-reported Hispanic/Latino with Hispanic/Latino surname) and specificity (99.14%—proportion of self-reported non-Hispanic/Latino with no Hispanic/Latino surname) [20]. Though NHIA classification does not capture women from Brazil some women in our sample identified as being born in Brazil. This study received IRB approval from all participating institutions. The final analytic sample included 5777 Latina/Hispanic invasive breast cancer cases (Figure 1), after excluding ductal carcinoma in situ (DCIS) cases (*n* = 11).

### 2.2. Categorization of Sociodemographic Characteristics

Sociodemographic characteristics including nativity, race, insurance status, nativity (i.e., birthplace), and area-level SES were provided by the NJSCR. Race was categorized as White, Black, Other, and Unknown [21]. “Other” included individuals identified as American Indian/Alaskan Native or Asian/Pacific Islander. Primary payor (i.e., insurance status) at diagnosis/treatment was defined as insured (private insurance, etc.), insured but not specified, Medicaid, uninsured, or unknown) [22]. Area-level SES was based on the Yost Index [23,24] calculated from census tract-level 2010 census data, categorized into quintiles. This composite index included: education index (weighted school years) [25], percent unemployed, percent working class, median household income, percent below 150% of poverty line, median house value, and median rent.

### 2.3. Clinicopathological Variables

BC subtypes were classified using NJSCR Site Specific Factor 16 (SSF16) data, which combines ER, PR, and HER2 status. Because cancer registries did not routinely collect HER2 data for incident breast cancer diagnosed before 1 January 2010, our prior research improved the capture of HER2 data for diagnosis years 2008 and 2009, which were validated using existing records for 2010–2013 diagnoses [26]. Our findings demonstrated 86.8% completeness of BC subtype in NJSCR [26]. Data on tumor grade (well differentiated, moderately differentiated, and poorly differentiated which included undifferentiated/anaplastic), tumor size (≤5 mm, >5–10 mm, >10–20 mm, >20–40 mm, and >40 mm), histology (ductal carcinoma, lobular carcinoma, both ductal and lobular carcinoma, mixed, and other) [26], and tumor stage (Stages I, II, III, IV) were also retrieved from NJSCR for each BC case included in the analysis.

### 2.4. Oncotype DX (ODX) Test Eligibility

Criteria for ODX testing was based on NCCN guidelines. Eligibility for cases diagnosed from 2008 to 2014 included HR+, HER2−, node-negative (N0), and tumor size >5 mm. Due to updated NCCN guidelines in 2015, eligibility for cases diagnosed from 2015 to 2017 included lymph node-positive (1–3 positive axillary lymph nodes), stages I–III, and any tumor size. BC cases were considered ineligible if they had missing data on any of the eligibility criteria or had a histologic subtype of mucinous or tubular adenocarcinoma (ICD-O-3: 8211, 8480, 8481), which are unfavorable for ODX testing. ODX test receipt and RS were determined using a data linkage between NJSCR and Genomic Health Institute (GHI) [4]. Based on the NCCN criteria, when a BC case that were eligible for ODX testing and did not receive the test, it was considered as “under-utilization” and when a BC case was ineligible for ODX testing but still received the test, it was categorized as “over-utilization”. The main focus of this paper is the under-utilization of ODX testing. Among cases that received the test, ODX RS was categorized as low risk (0–17), intermediate risk (18–30), and high risk (≥31), based on standard 21-gene assay cut-off points [4].

### 2.5. Statistical Analysis

Distributions of select sociodemographic and breast tumor clinicopathologic characteristics of all Latina/Hispanic women in New Jersey from 2008 to 2017 were described using means and standard deviations (mean ± SD) and frequency distributions [*n* (%)] (Appendix A). Among Latina/Hispanic cases who were eligible to receive ODX testing (*n* = 1916), distributions of select sociodemographic and breast tumor clinicopathologic features were compared by test receipt using Student’s t-tests and Chi-square tests. Trend tests were performed on ordinal variables. We performed an exploratory analysis to understand the distributions of ODX RS, which were described by means and standard (mean ± SD) as well as their range. ODX risk groups were described using frequency distributions [*n* (%)] among all women who received testing and separately by under-utilization vs. over-utilization of ODX testing. Student’s t-test were used to demonstrate differences in ODX recurrence scores, and Chi-square tests was used to examine differences among ODX risk groups. Kaplan–Meier survival curves are presented comparing ODX utilization and across risk groups for all women who received ODX testing where exit date was death or the end of the follow-up (31 December 2017). Survival end points included both all cause and BC-specific mortality that were collected by NJSCR.

Unadjusted odds ratios (OR) and adjusted odds ratios (AOR) of factors associated with ODX test receipt among eligible women were reported (with 95% confidence intervals [CI]) using multivariable logistic regression. Based on prior literature suggesting that SES factors and clinicopathologic features (e.g., age at diagnosis, tumor grade) were associated with ODX test receipt [10,13], we decided a priori to consider these factors as covariates in the adjusted model. Nativity was excluded from regression models due to the high (>50%) missing responses. All reported *p*-values are two-sided and *p* < 0.05 was considered statistically significant. Analyses were performed using R statistical software version 3.5.1.

## 3. Results

One-third (33.2%) of all Latina/Hispanic BC cases diagnosed between 2008 and 2017 in New Jersey were eligible to receive ODX testing (1916/5777). Among the eligible women (*n* = 1916), 60.9% (1167/1916) received the test. Overall, the proportion of all women receiving ODX testing increased each year during the study period and this trend was also observed among women who were eligible for ODX testing (Figure 2). 

As shown in Table 1, ODX test receipt among Latina/Hispanic women differed by sociodemographic (age at diagnosis, insurance status, area-level SES) and all tumor clinicopathologic characteristics with the exception of histology.

Approximately 73% (1167/1608) of all women who received an ODX test were eligible to receive ODX testing based on NCCN guidelines. The average ODX RS was lower among women who were eligible and received ODX testing compared to those who were ineligible to receive ODX testing but received testing (16.50 ± 8.98 vs. 18.63 ± 11.23, *p* < 0.001) (Table 2). The majority of women who were eligible and received ODX testing were categorized as low risk (63.2%) or intermediate risk (30.2%). We also observed a higher proportion classified as high risk among women who were ineligible for ODX testing compared to those who were eligible for testing (*p* < 0.001).

There were statistically significant differences in BC survival among women who received ODX testing (Figure 3), where the lowest survival probability was observed among women in the high-risk group (Log Rank *p* = 0.0001). The 5- and 10-year survival probabilities for women who were in the low risk ODX risk group were 98.2% and 88.9%, respectively. Similarly, women in the intermediate risk group had a 5- and 10-year survival of 94.2% and 86.9%, respectively. Those in the high risk group had a 5-year survival of 87.7% and at 10-years 72.2%, respectively.

We observed no statistically significant differences in survival among women who were ineligible for testing, but received testing compared to eligible women who received testing (log rank *p* = 0.610) (Figure 4). The 5- and 10-year survival probability for women who were eligible and received ODX testing were 96.5% and 87.3%, respectively, compared to 95.8% and 86.1% for women who were ineligible and received ODX testing.

Table 3 shows the multivariable logistic regression models for factors associated with ODX test receipt among eligible Latina/Hispanic women (*n* = 1916). In unadjusted analysis, older age was associated with lower odds of ODX test receipt (70–79 years: OR 0.61, 95% CI: 0.46, 0.81; ≥80 years: OR 0.14, 95% CI: 0.07, 0.30; *p*_trend_ < 0.001). Ethnic subgroup was not significantly associated with ODX test receipt. Women who were uninsured had 41% lower odds of receiving ODX testing compared to insured women. In comparison to residence in high SES areas, residence in lower SES areas had 42% lower odds of test receipt (OR = 0.58, 95% CI: 0.43, 0.77). Area-level SES did not achieve statistical significance for linear trend (*p*_trend_ = 0.329). In the adjusted model, older age (70–79 years: AOR 0.43, 95% CI: 0.31, 0.59; ≥80 years: AOR 0.08, 95% CI: 0.04, 0.18), being uninsured (AOR 0.58, 95% CI: 0.39, 0.86), residence in a low SES area (AOR 0.58, 95% CI: 0.42, 0.82), and more advanced tumor stage at diagnosis (Stage II: AOR 0.34, 95% CI: 0.22, 0.54; Stage III: AOR 0.06, 95% CI: 0.02, 0.22) were associated with lower odds of ODX test receipt. Conversely, moderately differentiated tumor grade (AOR 1.61, 95% CI: 1.22, 2.13) and larger tumor size were associated with increased odds of ODX test receipt (*p*_trend_ < 0.001).

## 4. Discussion

The results of this study demonstrated relatively high rates of ODX testing (60.9%) among eligible Latina/Hispanic women diagnosed with BC from 2008 to 2017 in New Jersey. This rate of ODX test receipt was higher than previously reported among Latina/Hispanic women across both Surveillance, Epidemiology, and End Results (SEER) Program and National Cancer Database (NCDB) registries [10,13,27,28]. The high proportion of ODX testing may be due to the increasing use of ODX in recent years as our sample of BC cases included those diagnosed through the end of 2017 where most studies collected data up to 2015. Prior to 2017, eligibility requirements expanded to include more tumor characteristics and insurances companies such as Medicare expanded coverage for [29]. These increasing trends have been reported for New Jersey Medicare beneficiaries, showing they are more likely to receive ODX testing compared to individuals in other regions of the United States [30].

While ODX testing is important to determine risk of recurrence, it is particularly informative as to whether or not adjuvant chemotherapy would be beneficial treatment option [31,32]. One study demonstrated that, in the absence of ODX testing, non-Hispanic Black women and Latina/Hispanic women were more likely to receive chemotherapy, thereby exposing them unnecessarily to greater risk of treatment toxicity [27] and potentially lower quality of life post-acute cancer treatment. Our results support these previous findings where women who did not receive ODX testing were more likely to have reported chemotherapy. Moreover, survival curves differentiating the three risk groups show that women who had low ODX RS had the highest survival probability compared to those with intermediate and high RS. Thereby demonstrating the importance of ODX testing to ensure that women with low RS do not receive unnecessary adjuvant chemotherapy. This accords with Schwedhelm et al. [33], where women who were categorized as a low ODX risk group had decreasing rates of chemotherapy use from 2010 to 2016.

We hypothesized that both insurance status at diagnosis and area-level SES is associated with odds of ODX test receipt. We found that women who were uninsured and those residing in areas characterized as low SES had the lowest odds of ODX test receipt. According to Centers for Medicaid and Medicare Services (CMS), as of 2015 the cost of getting an ODX test was around $3400 [34], which may be cost-prohibitive for uninsured/underinsured individuals and others who are on the lower end of the individual-level SES spectrum. Interestingly, while we did not achieve statistical significance, the effect estimate for Medicaid enrolled women showed increased odds of receiving ODX, which was not observed in a study examining ODX utilization from various SEER registries [9]. Future analyses including larger samples of Latina/Hispanic BC cases from broader geographic areas are warranted to clarify the relationships between insurance status and ODX test receipt.

It is notable that age at diagnosis was strongly associated with ODX test receipt, with significantly lower odds of ODX testing among older Latina/Hispanic women. This finding is fairly consistent with other studies that show an inverse association between age and ODX test receipt [10,13,27,35,36], and supports the idea that benefits of adjuvant chemotherapy may not increase life expectancy among older women, particularly those with multiple comorbidities [37]. However, this finding was of concern because older women with higher ODX RS tend to have higher 5-year BC specific-mortality compared to women of the same age with lower ODX RS (ranging from 10.4% to almost 22%) [4]. Therefore, it is crucial to ensure that the aging Latina/Hispanic population receives appropriate ODX testing to tailor treatment and ultimately reduce BC mortality rates.

We also hypothesized that there would be an association between ODX test receipt and ethnic subgroup and/or nativity (US-born vs. non-US born) given the growing literature demonstrating cancer health disparities associated with nativity [12,38,39,40,41,42]. We also considered these variables to be important because they might be a proxy for environmental exposures or other factors from earlier in the life course, and thereby might contribute to BC outcomes [43,44,45]. We found no association between ethnic subgroup with ODX test receipt and given the large percentage (>50%) of missing data on nativity from the NJSCR, it was not included in the regression models. Availability of nativity information is a known limitation of population-based cancer registries [45]. This study highlights the need for more complete information on birthplace in population-based cancer registries to facilitate generation of disaggregated cancer surveillance data to facilitate understanding and addressing cancer inequities within racial/ethnic subgroups.

We observed a statistically significant, positive association with tumor size and ODX test receipt, which was reported in an analysis of data from 14 SEER registries [4]. However, our effect estimates were much higher than previously reported and the confidence intervals were wide, indicating that our finding might be subject to random error and should be interpreted cautiously. We conducted sensitivity analysis including year of diagnosis in the adjusted regression model since ODX guidelines did not include ≤5 mm tumor size prior to 2015 and we did not observe a significant change in the effect estimates obtained (data not shown).

In terms of ODX RS, more than half of the study sample was classified as low risk, which was a promising finding as low recurrence risk reduces the necessity for adjuvant chemotherapy [46]. While we found differences in the RS among women who were eligible and ineligible for ODX testing, we did not observe any differences in survival in our sample. Moreover, among all women who received the ODX test higher survival was observed among those low or intermediate RS, consistent prior data [4]. We also analyzed ODX risk groups using the TAILORx cut points [47] and the results remained materially unchanged (data not shown). Nonetheless, these findings must be interpreted cautiously as Kaplan Meier survival curves were conducted as part of exploratory analyses stemming from statistically significant differences in ODX RS among women who were eligible for ODX testing and received testing compared to women who were ineligible and received testing.

This study has some limitations that should be considered in the interpretation of our findings. While ODX test receipt is reportable to cancer registries, there is no way to confirm that every test was reported. In other words, we cannot rule out the possibility of incomplete data capture for ODX tests done among women included in our study sample. However, while a data linkage with GHI was not done prior to 2011, data on ODX testing among BC cases diagnosed from 2011 to 2015 in New Jersey was found to be relatively complete (96%) [48], which minimized this concern. Another limitation relates to the generalizability of our findings. In New Jersey, Latinos/Hispanics account for approximately 18% of the population, making it the state with the seventh largest Hispanic/Latino population in the nation [49], and there is high within-group diversity in terms of ethnic subgroup and nativity. Therefore, the associations found in this study may not be observed in other states where distributions of Latino/Hispanic populations are significantly different. In addition, we do not have individual-level SES data in order to examine the association with ODX testing and use area-level SES instead [23].

Our study also had some notable strengths. This is the first study to specifically examine ODX test receipt among Latina/Hispanic women with BC in New Jersey and to explore factors associated with the receipt of ODX testing among eligible women. We further investigated ODX test receipt by Hispanic/Latino subgroup, demonstrating the heterogeneity within Latino/Hispanic populations. These novel data have not been reported in other studies focusing on ethnic/racial disparities in ODX test receipt, where many of these studies combined cases across cancer registries [8,9,10,13,27,50]. While these studies are statistically powered to detect associations and are generalizable, our findings specifically focus on inequity of health services utilization among Latina/Hispanic women in New Jersey. Finally, the NJSCR performed a linkage with GHI for BC cases diagnosed from 2011 to 2015 and demonstrated a 96% completeness rate for ODX testing data for this timeframe. Ergo, demonstrating the robustness of the data included in this study.

## 5. Conclusions

Among Latina/Hispanic women diagnosed with BC in New Jersey from 2008 to 2017, there was relatively high ODX test receipt among eligible women (60.9%). Our findings suggest that age at diagnosis, insurance status, area-level SES, and tumor grade are important correlates of ODX test receipt. These findings highlight the clinical importance of further investigating factors that contribute to receipt of optimal treatment among Latina/Hispanic women diagnosed with BC, which is essential to understanding and improving BC outcomes and minimizing short- and long-term treatment effects from unnecessary chemotherapy in this group.

## Figures and Tables

**Figure 1 ijerph-18-05116-f001:**
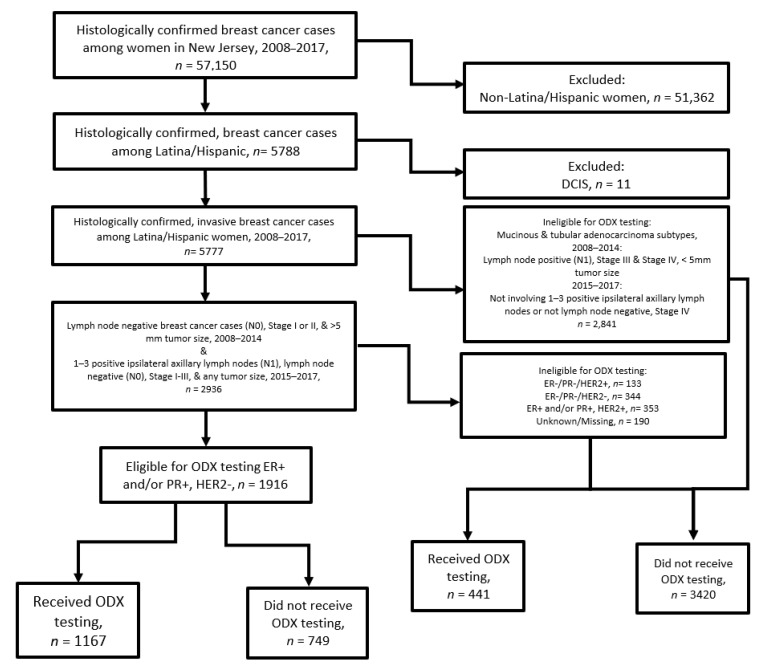
Flow diagram depicting assembly of the analytic sample.

**Figure 2 ijerph-18-05116-f002:**
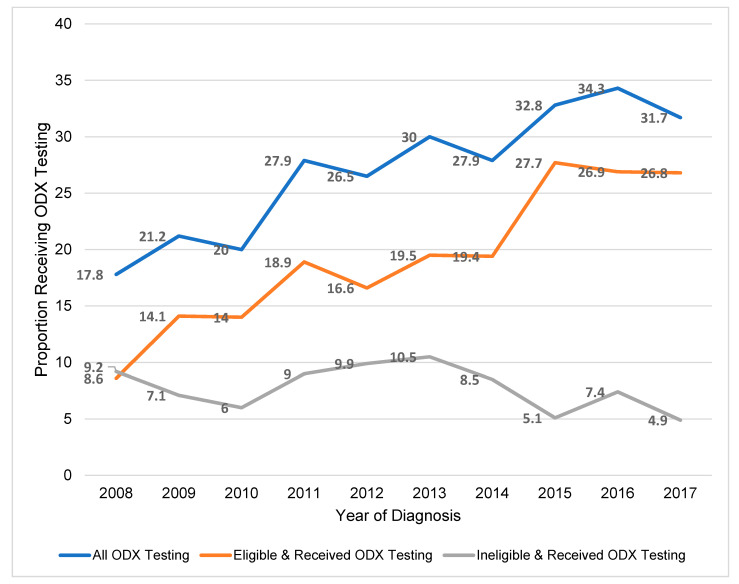
Proportion of all Latina/Hispanic women in New Jersey diagnosed with early invasive BC who received ODX testing.

**Figure 3 ijerph-18-05116-f003:**
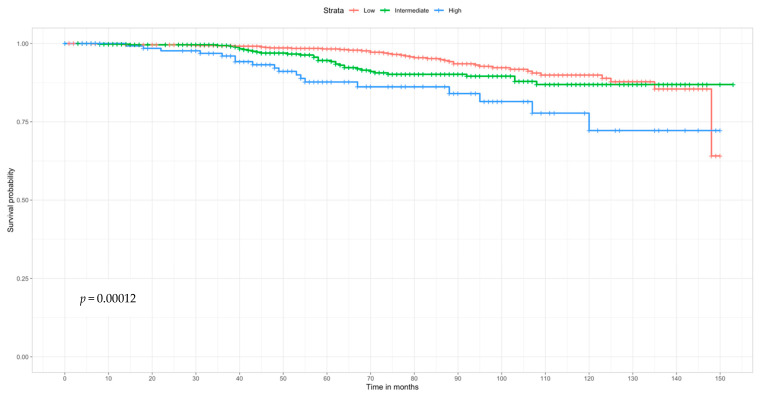
Kaplan–Meier curve of all Latina/Hispanic women who received ODX testing by risk group (*n* = 1608).

**Figure 4 ijerph-18-05116-f004:**
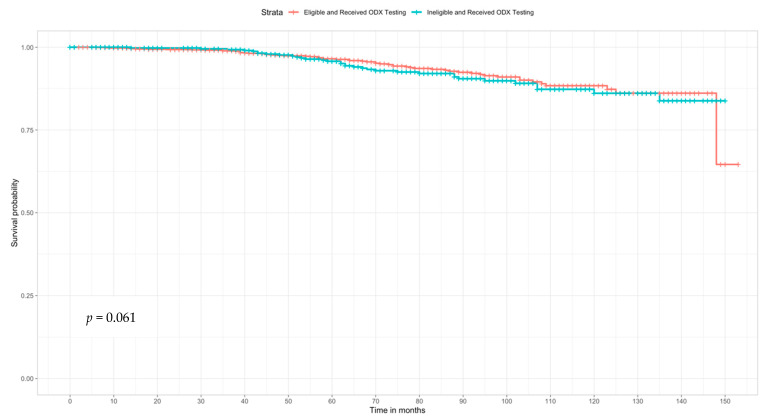
Kaplan–Meier curve of all Latina/Hispanic women who received ODX testing by eligibility (*n* = 1608).

**Table 1 ijerph-18-05116-t001:** Oncotype DX test receipt by demographic characteristics among Latina/Hispanic women diagnosed with breast cancer who were eligible for the test (*n* = 1916), New Jersey, 2008–2017.

	Received ODX Test (*n* = 1167)	Did Not Receive ODX Test (*n* = 749)	*p*
Sociodemographic characteristics			
Age at Diagnosis, y			<0.001
Mean ± SD	56.33 ± 11.22	58.59 ± 13.71	
Range	24.0–88.0	20.0–91.0	
Age at Diagnosis			<0.001
<50	360 (30.8)	214 (28.6)	
50–59	343 (29.4)	182 (24.3)	
60–69	299 (25.6)	164 (21.9)	
70–79	156 (13.4)	151 (20.1)	
≥80	9 (0.8)	38 (5.1)	
Race			0.829
White	1046 (89.6)	681 (90.9)	
Black	53 (4.6)	32 (4.3)	
Other ^a^	62 (5.3)	36 (4.8)	
Missing/Unknown	6 (0.5)	0 (0)	
Ethnic Subgroup ^b^			0.284
Mexican	44 (3.8)	32 (4.3)	
Puerto Rican	192 (16.5)	144 (19.2)	
Cuban	67 (5.7)	32 (4.3)	
South or Central American	223 (19.1)	160 (21.3)	
Spanish/Hispanic/Latino NOS	361 (30.9)	212 (28.3)	
Dominican	59 (5.1)	41 (5.5)	
Other	221 (18.9)	128 (17.1)	
Nativity			0.312
U.S. Born	139 (11.9)	79 (10.6)	
Non-US-born	424 (36.3)	287 (38.3)	
Unknown/Missing	604 (51.8)	383 (51.1)	
Insurance			0.018
Insured	740 (63.4)	440 (58.7)	
Medicaid	178 (15.3)	107 (14.3)	
Uninsured	77 (6.6)	77 (10.3)	
Insured, but Not Specified	140 (12.0)	98 (13.1)	
Missing/Unknown	32 (2.7)	27(3.6)	
Area-Based Composite Socioeconomic Status ^c^			0.001
Low	196 (16.8)	169 (22.5)	
Low-middle	219 (18.7)	140 (18.7)	
Middle	198 (17.0)	125 (16.7)	
Middle-high	239 (20.5)	160 (21.4)	
High	299 (25.6)	149 (19.9)	
Unknown/Missing	16 (1.4)	6 (0.8)	
Clinical characteristics			
Histologic Type			0.446
Ductal	886 (75.9)	556 (74.2)	
Lobular	118 (10.1)	69 (9.2)	
Both Lobular and Ductal	91 (7.8)	62 (8.3)	
Mixed	44 (3.8)	36 (4.8)	
Other/Unknown	28 (2.4)	26 (3.5)	
Tumor Stage			<0.001
Stage I	849 (72.8)	444 (59.3)	
Stage II	312 (26.7)	283 (37.8)	
Stage III	6 (0.5)	22 (2.9)	
Tumor Grade			<0.001
Well Differentiated	211 (18.1)	150 (20.0)	
Moderately Differentiated	702 (60.2)	358 (47.8)	
Poorly Differentiated ^d^	213 (18.2)	203 (27.1)	
Missing/Unknown	41 (3.5)	38 (5.1)	
Tumor Size			<0.001
≤5 mm	27 (2.3)	79 (10.5)	
>5–10 mm	294 (25.2)	157 (21.0)	
>10–20 mm	574 (49.2)	267 (35.6)	
>20–40 mm	243 (20.8)	187 (25.0)	
>40 mm	29 (2.5)	59 (7.9)	
Tumor Subtype			<0.001
ER−/PR+/HER2−	2 (0.2)	24 (3.2)	
ER+/PR−/HER2−	103 (8.8)	106 (14.2)	
ER+/PR+/HER2−	1062 (91.0)	619 (82.6)	
Reported Chemotherapy			<0.001
Yes	276 (23.7)	295 (39.4)	
No	852 (73.0)	425 (56.7)	
Unknown	39 (3.3)	29 (3.9)	

^a^ ‘Other’ race includes American Indian/AK Native and Asian/Pacific Islander. ^b^ Ethnic subgroup was recoded using the indirect identification from the NAACCR Hispanic/Latino Identification Algorithm [NHIA v.2.2.1] of “surname match only” (sensitivity = 84.37% and specificity = 99.14%). ^c^ Area-based composite socioeconomic status was based on the Yost Index using US 2010. Census Tract data. ^d^ ‘Poorly differentiated’ includes undifferentiated/anaplastic tumors.

**Table 2 ijerph-18-05116-t002:** Oncotype DX recurrence risk score and risk group among Latina/Hispanic breast cancer cases who received the test (*n* = 1608), New Jersey, 2008–2017.

Variable	Received ODX Test (*n* = 1608)	Eligible & Received (*n* = 1167)	Ineligible & Received (*n* = 441)	*p*
Oncotype DX Recurrence Score				
Mean ± SD	17.09 ± 9.69	16.50 ± 8.98	18.63 ± 11.23	<0.001
Range	0–63.0	0–63.0	0–62.0	
Oncotype DX Risk Group ^a^				
Low Risk	565 (54.5)	737 (63.2)	253 (57.4)	<0.001
Intermediate Risk	277 (26.7)	353 (30.2)	130 (29.5)	
High Risk	83 (8.0)	77 (6.6)	58 (13.2)	

^a^ ODX risk groups categorized as low risk 0—17, intermediate risk 18—30, and high risk ≥31.

**Table 3 ijerph-18-05116-t003:** Logistic regression analysis of Oncotype Dx test receipt among Latina/Hispanic breast cancer cases who were eligible for the test in New Jersey, 2008–2017.

	Unadjusted OR			Adjusted OR ^a^		
Covariate	OR	95% CI	*p* _trend_	AOR	95% CI	*p* _trend_
Age at Diagnosis						
<50	1.00	(Ref)	<0.001	1.00	(Ref)	<0.001
50–59	1.12	0.88, 1.43		0.97	0.73, 1.28	
60–69	1.08	0.84, 1.40		0.89	0.67, 1.19	
70–79	0.61	0.46, 0.81		0.43	0.31, 0.59	
≥80	0.14	0.07, 0.30		0.08	0.04, 0.18	
Ethnic Subgroup						
Spanish/Hispanic/Latino NOS	1.00	(Ref)		1.00	(Ref)	
Cuban	1.23	0.78, 1.94		1.60	0.95, 2.70	
Dominican	0.85	0.55, 1.30		0.97	0.60, 1.57	
Mexican	0.81	0.50, 1.31		0.82	0.47, 1.44	
Other	1.01	0.77, 1.34		0.87	0.63, 1.19	
Puerto Rican	0.78	0.59, 1.03		0.76	0.55, 1.04	
South or Central American	0.82	0.63, 1.07		0.93	0.69, 1.26	
*Insurance*						
Insured	1.00	(Ref)		1.00	(Ref)	
Insured, NOS	0.85	0.64, 1.13		0.92	0.67, 1.27	
Medicaid	0.99	0.76, 1.29		1.32	0.97, 1.81	
Uninsured	0.59	0.42, 0.83		0.58	0.39, 0.86	
Area-Based Composite Socioeconomic Status						
High	1.00	(Ref)	0.329	1.00	(Ref)	0.472
Middle–High	0.74	0.56, 0.99		0.76	0.56, 1.05	
Middle	0.79	0.59, 1.06		0.77	0.55, 1.08	
Low–Middle	0.78	0.58, 1.04		0.82	0.59, 1.14	
Low	0.58	0.43, 0.77		0.58	0.42, 0.82	
Tumor Stage						
Stage I	1.00	(Ref)		1.00	(Ref)	
Stage II	0.58	0.47, 0.70		0.34	0.22, 0.54	
Stage III	0.14	0.06, 0.35		0.06	0.02, 0.22	
Tumor Grade						
Well Differentiated	1.00	(Ref)		1.00	(Ref)	
Moderately Differentiated	1.39	1.09, 1.78		1.61	1.22, 2.13	
Poorly Differentiated	0.75	0.56, 0.99		0.82	0.59, 1.12	
Tumor Size						
≤5 mm	1.00	(Ref)	<0.001	1.00	(Ref)	<0.001
>5–10 mm	5.48	3.40, 8.84		6.03	3.61, 10.09	
>10–20 mm	6.29	3.97, 9.97		7.64	4.64, 12.56	
>20–40 mm	3.80	2.36, 6.12		12.30	6.27, 24.03	
>40 mm	1.44	0.77, 2.68		5.24	2.31, 11.89	

^a^ Adjusted odds ratios (AOR) and 95% confidence intervals (CI) were calculated using logistic regression models with adjustment for all variables in the model.

## Data Availability

The datasets generated during and/or analyzed in the current study are available from the corresponding author on reasonable request.

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
