# Peer review of "Oncotype DX Test Receipt among Latina/Hispanic Women with Early Invasive Breast Cancer in New Jersey: A Registry-Based Study"

_ijerph, 2021, doi:10.3390/ijerph18105116_

Round 1

Reviewer 1 Report

This manuscript addresses the use of Oncotype Dx testing among Latina/Hispanic women with early stage breast cancer in New Jersey. They examine overall rates of testing, rates among those eligible/ineligible for testing and variables associated with these differences. 

Consideration of those tested who were otherwise ineligible is an important contribution. More information characterizing this group would be valuable. 

-Reference 4 should be reconsidered. 

Also, a number of sensitivity analyses could be done:

-Is the numerical value of ER/PR positivity available? If so, these could be tested as continuous scores. 

-Were the test category cutoffs used reflecting the test validation cutoffs or the Tailorx trial? Given these data come partially from before the trial was complete, both could be considered. 

Discussion: Further discussion of recent work by Hoskins et al (PMID: 33475714) on how testing performs in African American women could be added. This could lead to consideration of test performance among Latinas.

Reviewer 2 Report

This manuscript examines factors associated with ODX testing among Hispanic women using data from the New Jersey State Cancer Registry. The manuscript is generally well-written and the results are interpreted soundly. I have a few suggestions for areas in need of clarification or  improvement, enumerated below.

  1. The numbers reported in Figure 2 indicate that about 30% of Hispanic patients received ODX testing but the text says 60.9% of eligible Hispanic women received ODX. I am not following what the difference between these two statistics is or why there is a discrepancy.
  2. Consider reporting 5- or 10-year survival in Figure 3. The rationale for showing the Kaplan Meier curves comparing eligible vs ineligible patients who received ODX is unclear. Did the authors consider analyzing differences between those who are eligible and received ODX and eligible but did not receive ODX? Kaplan Meier curves on their own are not the best representation of survival differences. A Cox Proportional Hazards model would better model differences in survival adjusting for other relevant patient factors.
  3. There have been several studies examining patient characteristics associated with ODX testing using national (NCDB), regional (SEER), and state registries. While larger datasets are more powered to detect associations, they may miss local/regional disparities that are important to uncover. There is an opportunity for the authors to explain benefits and limitations to each data source in the discussion. There have been other studies using data from single state cancer registries (including but not limited to Davis et al and Schwedhelm et al) to study variation in ODX testing and chemotherapy that could be referenced. Regional variation in ODX testing has been shown using nationwide Medicare data (Lynch et al), indicating that some regions are generally more likely to prescribe the test than others, and the underlying propensity for the region to prescribe ODX may impact the utilization patterns among different sub-populations in that region. The focus of a specific population (i.e., Hispanic women) in the state of NJ may not be completely generalizable but these results will likely be impactful to understanding equity within that state and state-level policy, representing an important contribution.

Davis BA et al. Racial and ethnic disparities in Oncotype DX test receipt in a statewide population-based study. JNCCN. 2017; 19(3).

Schwedhelm TM et al. Patient and physician factors associated with Oncotype DX and adjuvant chemotherapy utilization for breast cancer patients in New Hampshire, 2010-2016. BMC Cancer. 2020; 20(847).

Lynch JA et al. 21-Gene recurrence score testing among Medicare beneficiaries with breast cancer in 2010-2013. Genetics in Medicine. 2017; 19:1134-1143.

Round 2

Reviewer 2 Report

The authors have adequately responded to all comments.